# *USH2A*-Related Retinitis Pigmentosa: Staging of Disease Severity and Morpho-Functional Studies

**DOI:** 10.3390/diagnostics11020213

**Published:** 2021-02-01

**Authors:** Benedetto Falsini, Giorgio Placidi, Elisa De Siena, Maria Cristina Savastano, Angelo Maria Minnella, Martina Maceroni, Giulia Midena, Lucia Ziccardi, Vincenzo Parisi, Matteo Bertelli, Paolo Enrico Maltese, Pietro Chiurazzi, Stanislao Rizzo

**Affiliations:** 1Università Cattolica del Sacro Cuore, 00168 Rome, Italy; benedetto.falsini@unicatt.it (B.F.); giorgioplacidi@libero.it (G.P.); elisa.desiena@yahoo.com (E.D.S.); mariacristina.savastano@unicatt.it (M.C.S.); angelomaria.minnella@unicatt.it (A.M.M.); giulia.midena@gmail.com (G.M.); pietro.chiurazzi@unicatt.it (P.C.); stanislao.rizzo@unicatt.it (S.R.); 2UOC Oftalmologia, Fondazione Policlinico Universitario A. Gemelli IRCCS, 00168 Rome, Italy; 3Fondazione GB Bietti per l’Oftalmologia, IRCCS, 00184 Rome, Italy; lucia.ziccardi@fondazionebietti.it (L.Z.); vincenzo.parisi@fondazionebietti.it (V.P.); 4MAGI’S LAB, 38068 Rovereto, Italy; matteo.bertelli@assomagi.org (M.B.); paolo.maltese@assomagi.org (P.E.M.); 5MAGI EUREGIO, 39100 Bolzano, Italy; 6UOC Genetica Medica, Fondazione Policlinico Universitario A. Gemelli IRCCS, 00168 Rome, Italy

**Keywords:** Usher syndrome, Usher *2A* gene, retinitis pigmentosa, staging, electroretinogram, retinal pigment epithelium (RPE) and outer retina atrophy (RORA), sub-RPE illumination (SRI)

## Abstract

Usher syndrome type 2A (*USH2A*) is a genetic disease characterized by bilateral neuro-sensory hypoacusia and retinitis pigmentosa (RP). While several methods, including electroretinogram (ERG), describe retinal function in *USH2A* patients, structural alterations can be assessed by optical coherence tomography (OCT). According to a recent collaborative study, RP can be staged considering visual acuity, visual field area and ellipsoid zone (EZ) width. The aim of this study was to retrospectively determine RP stage in a cohort of patients with *USH2A* gene variants and to correlate the results with age, as well as additional functional and morphological parameters. In 26 patients with established *USH2A* genotype, RP was staged according to recent international standards. The cumulative staging score was correlated with patients’ age, amplitude of full-field and focal flicker ERGs, and the OCT-measured area of sub-Retinal Pigment Epithelium (RPE) illumination (SRI). RP cumulative score (CS) was positively correlated (r = 0.6) with age. CS was also negatively correlated (rho = −0.7) with log10 ERG amplitudes and positively correlated (r = 0.5) with SRI. In *USH2A* patients, RP severity score is correlated with age and additional morpho-functional parameters not included in the international staging system and can reliably predict their abnormality at different stages of disease.

## 1. Introduction

Usher syndrome type 2 (USH2) is a genetic disease characterized by congenital or early bilateral neuro-sensory hypoacusia and later onset of night blindness, visual acuity loss and constricted visual field due to co-occurring retinitis pigmentosa (RP). USH2 is the most common cause of deaf-blindness and the most frequent form of recessive RP, comprising 18% of all RP [1].

USH2 is genetically heterogeneous [2,3,4,5]. There are three *USH2* genes encoding *USH2A* (MIM 276901), *ADGRV1* (MIM 605472) and *WHRN* (MIM 607928). USH2 is characterized by a lack of vestibular deficits, mild to generally moderate degrees of congenital neuro-sensory hearing loss and RP onset usually in the second decade of life. In USH2 patients, RP symptoms typically start with a concentric visual field loss, while visual acuity can be relatively well-preserved for a long time [6,7]. Progression of visual field loss may occur at a much faster rate than losses in visual acuity or cone-mediated central light-adapted perimetry [6,7,8].

Emerging treatments for *USH2A-*related retinal degeneration are currently under study. One strategy, based on genome editing [9], has shown efficacy for correction of the most prevalent *USH2A* variant in patient-derived induced pluripotent stem cells. The other, based on antisense oligonucleotides [10], has been effective for in vitro correction of *USH2*A variants in patient-derived fibroblasts. Both approaches show promise for clinical efficacy.

In light of future applications of these therapeutic approaches, it is becoming important to quantify the severity grade and progression of the disease. Ganzfeld flash electroretinogram (ERG), by using specialized microvolt techniques [11] and focal electroretinograms [12], has been used to evaluate the progression of retinal dysfunction in *USH2A*. Spectral domain-optical coherence tomography (SD-OCT) provides objective morphological parameters to quantify the progression stage of RP-related to *USH2A*. The ellipsoid zone (EZ) width has been largely employed to determine the severity of central retinal damage. Additional parameters to quantify macular damage are now standardized and employed, such as sub-Retinal Pigment Epithelium (RPE) illumination (SRI) [13].

A recent collaborative study [14] based on a cross-sectional analysis of a large cohort of patients with non-syndromic RP introduced a new classification system to stage the severity of the disease. A scoring criterion was developed considering Early Treatment Diabetic Retinopathy Study (ETDRS) visual acuity, Goldmann visual field area and EZ width, and by assigning a score from 0 to 5 to each variable. Normal values correspond to a score of 0 (85 ETDRS letters or better for visual acuity, 120° or better for visual field diameter and 30° or better for EZ width), whereas a score of 1–5 is assigned based on the population quintiles [14]. The cumulative score (CS, from 0 to 15) is represented by the sum of all scores and it is used to determine the severity grade from 0 to 5 [14].

The aim of the present study was to determine the disease severity stage using the classification of Iftikhar et al. [14], in a cohort of patients with syndromic and non-syndromic RP due to variants in the *USH2A* gene, and to correlate the results with age and additional morpho-functional parameters.

## 2. Materials and Methods

This retrospective study was approved by the Ethics Committee/Institutional Review Board of the Catholic University of Rome, Italy. This research adhered to the tenets of the Declaration of Helsinki and informed consent was obtained from all patients, after full and detailed explanation of the goals and procedures of the study. All the clinical, imaging and electrophysiological data reported in this study were retrospectively re-evaluated.

### 2.1. Subjects

This is a cross-sectional study based on data from 23 patients with syndromic or non-syndromic RP caused by *USH2A* variants and clinically followed at the Center for Inherited Retinal Degenerations of Fondazione Policlinico Gemelli.

Inclusion criteria were clinical and genetic diagnosis of syndromic or non-syndromic RP likely determined by causative variants in *USH2A* gene, absence of concomitant ocular or systemic disorders, sufficiently clear optical media for performing retinal imaging and good cooperation in psychophysical testing.

Exclusion criteria were phenotype or genotype not referred to *USH2A* variants, concomitant ocular or systemic disorders, cataract or vitreous opacities that precluded performing reliable retinal imaging and poor cooperation in psychophysical testing.

The mean age of the included patients was 42 years (±16 standard deviation (SD)). All patients had causative variants in the *USH2A* gene, confirmed, when possible, by segregation analysis: seven patients were homozygous, and the remaining were compound heterozygous. Clinically, 20 patients had syndromic RP associated with neuro-sensory hearing loss and three patients had non-syndromic RP.

### 2.2. Data Acquisition

Each enrolled patient underwent a full ophthalmologic examination, including best corrected visual acuity (BCVA) measured with ETDRS charts, Goldmann visual field using the V/4 e and III/4e test targets, direct and indirect ophthalmoscopy, scotopic and photopic full-field ERG, 30 Hz flicker ERG with the assessment of response variability and signal-to-noise ratio (S/N) (see below), focal macular electroretinography according to a published technique [15] and SD-OCT with measurement of the EZ extension and SRI.

Clinical staging of disease severity was performed according to Iftikhar et al. [14]. Specifically, each of the clinical parameters of interest, BCVA, EZ extension and Goldmann visual field (V/4e target), were assigned a score following the published protocol.

### 2.3. Electroretinogram Assessment

Ganzfeld ERGs were recorded by skin electrodes according to a published method [16]. In addition to the standard ISCEV stimuli, ganzfeld flicker ERGs were obtained with a special protocol able to provide information about response variability and S/N ratio. First, response underwent an off-line discrete Fourier analysis in order to isolate the fundamental 1F component at 30 Hz, whose amplitude and phase were measured. Second, 4 blocks of ERG responses, each being the average of 150 events, were collected. Fourier analysis was performed on the total grand average and on every block average [11,17]. Third, the noise response was estimated by summing up odd and even events for each average block and for the total average. The total average noise of the four blocks was then Fourier-analyzed in order to isolate the 1F component at 30 Hz. The amplitude of this component was taken as the noise estimate.

### 2.4. Spectral Domain-Optical Coherence Tomography Assessment

SD-OCT was performed using Zeiss Cirrus 5000-HD-OCT Angioplex, sw version 10.0, (CarlZeiss, Meditec, Inc., Dublin, CA, USA). A High-Definition 5-Line Raster and a macular map (6 × 6 mm Macular Cube 512 × 128) were acquired. Extent of the preserved EZ line was manually measured with calipers in horizontal and vertical scans, centered at the fovea. On OCT line scans, two boundaries were identified: the proximal edge of the retinal pigment epithelium (pRPE) located adjacent to the photoreceptor outer segments and the EZ band. For all scans, the nasal and temporal edges of the EZ band were defined as the locations where the EZ band met the pRPE. The width of the EZ band was defined as the distance between these two locations. The Advanced RPE analysis software was used to automatically determine areas of SRI (mm^2^) for increased light penetration through atrophic OR, RPE and choriocapillaris, by means of the sub-RPE slab tools. The SRI identifies bright areas of increased light transmission beneath the RPE, indicating RPE atrophy, averaged over a circular area of 5 mm around the fovea by the automated OCT software.

### 2.5. Statistical Analysis

We analyzed both right and left eyes. In the study, we considered only the results from the right eyes. The data were evaluated for their distribution. Linear regression, parametric (Pearson’s) and non-parametric (Spearman’s) correlation analyses were employed according to the data distribution of the different parameters. For non-continuous ordinal variables such as cumulative severity score, and given the non-Gaussian data distribution, the non-parametric Spearman rank order correlation analysis was considered more appropriate than Pearson’s correlation.

ERG amplitude data were normalized to Log10 in order to best approximate normal distribution. In all the analyses, a *p* < 0.05 was considered as statistically significant.

## 3. Results

Genetic, demographic and clinical data are reported in detail for each patient in Table 1, Table 2 and Table 3, respectively.

According to the published staging criteria [14], eight patients were classified as mild (grade 0–2), five as moderate (grade 3) and ten as severe (grade 4–5).

We analyzed both right and left eyes and the results from the statistical analyses were substantially similar. In addition, data from right and left eyes were highly correlated (CS inter-eye correlation: r = 0.98). In the study, we considered only the results from the right eyes in order to not overestimate the *p*-values.

The mean SRI area within the 5 mm circle was 1.5 mm^2^ (range 0–9.8), whereas the mean SRI fovea distance was 1.3 mm (range 0–3.6). The mean focal ERG amplitude was 0.45 μV (0.1–1.99; normal range: 1.6–2.8) and the mean amplitude obtained using 30 Hz Ganzfeld flicker was 0.86 μV (0.05–2.79; Normal range: 20–60). Details on SRI analysis and electrophysiological data are listed in Table 2 and Table 3.

In Figure 1, images of infrared (IR), fundus autofluorescence (FAF), OCT B scan, sub-RPE platform and RPE outline of two patients with mild RP (Figure 1a) and severe (Figure 1b) RP are shown. It can be noted that, in mild RP, the B scan image shows the perifoveal loss of the outer retinal layer. The central preservation of the EZ corresponds to the internal edges of the hyper-autofluorescent ring visible on FAF. In severe RP, the B scan image shows a profound loss of photoreceptor outer segments, with central loss of the RPE, corresponding to hypo-autofluorescent areas on FAF. This picture is associated with changes in the sub-RPE platform (areas of increased SRI) within the 5 mm circle outlined in white. The software provides an automated measure of the SRI area (the sum of areas outlined in yellow in the RPE outline).

Figure 2 shows the CS from the right eye of each patient plotted as a function of age. Linear regression analysis showed that the score increases linearly with age, with a slope of 0.18/year. The r value is 0.65 (*p* < 0.001).

In Figure 3, the Focal Electroretinogram (FERG) Log 10 amplitudes are plotted as a function of the CS. It can be noted that, in our patients, disease severity stage, in terms of CS (0–15), was negatively correlated (Spearman’s) with Log 10 FERG amplitudes (rho = −0.72, *p* < 0.0001).

Figure 4 shows Log10 1F cone flicker microvolt ERG amplitude as a function of CS (Spearman’s, rho = −0.58; *p* = 0.004).

Figure 5a,b shows the results of correlation analysis between SRI and CS (Figure 5a) and EZ score (Figure 5b). SRI showed a positive correlation (Spearman’s) with both CS (with rho = 0.53; *p* = 0.01) and the EZ score (rho = 0.54, *p* = 0.007).

The FERG log10 amplitudes and full-field ERG were positively correlated with EZ score (FERG r = −0.75, *p* < 0.0001; full-field sub-microvolt ERG, r = −0.57, *p* < 0.0001).

No correlation was found between SRI and FERG and cone microvolt ERGs (*p* = not significant, ns). Table 4 summarizes the results of the statistical analyses.

## 4. Discussion

The present study was designed to evaluate, in *USH2A*-related retinal degeneration, the progression stage of disease, and to correlate this stage with age and additional morpho-functional parameters not included in the model of the staging scheme proposed by Iftikhar et al. [14]. We calculated this recently published staging scheme in patients with *USH2A* variants and RP. The results showed that both the cumulative staging score and staging grade were positively correlated with age of patients and were also significantly correlated with the additional morpho-functional parameters.

The correlation with age may be relevant for predicting the natural history of disease and for determining the best temporal window for therapeutic strategies. The linear regression between the disease score and age may roughly predict the stage of disease based on the age of a patient. Based on linear statistical regression, the score increases fewer than 2 points every 10 years. We suggest that for USH2A patients, the age of onset tends to be quite homogeneous. Therefore, the cumulative severity score results to be positively correlated with patient age. Additional longitudinal studies to correlate the disease progression between *USH2A* stage and age could have important clinical implications for understanding the time course of disease.

By looking at Table 2, there is an almost complete agreement between the stage grade of the right and left eye of each patient. All enrolled patients showed the same stage of disease in both eyes, except for the patients 6 and 8, in which the transition from one grade to another is determined by a difference in the cumulative staging score value equal to 2 and 1 points, respectively. Our findings also mean that we could apply a therapeutic intervention in one eye and use the contralateral eye as the internal control for each patient.

The linear correlation between the stage of disease and the amplitude loss of the first harmonic in the microvolt ERG and FERG, indicates that the residual cone photoreceptor function tends to decrease until reaching values poorly differentiable from noise level. Ganzfeld flash ERG has been considered for a long time as a gold standard for staging RP severity. However, in *USH2A* syndromic and non-syndromic RP, the ERG has been of limited use since its amplitude rapidly drops to undetectable levels during the course of disease. More advanced techniques such as the cycle-by-cycle ERG [11], recording retinal signals at microvolt or sub-microvolt levels, can be clinically used to stage and monitor the progression of retinal degeneration. Indeed, cone microvolt ERGs were able to model the decay of retinal function in *USH2A* [11]. Focal electroretinogram (FERG) has been found to reliably describe the time course of central retinal dysfunction in a longitudinal study with a long-term follow-up in *USH2A* patients [12]. FERG has provided evidence of early loss of cone function before the onset of visual acuity and visual field loss. The present results indicate that, for stages 0 to 3, ERG amplitude can be recorded, but when stages 4 or 5 are reached, the ERG amplitude is so low that it cannot be reliably distinguished from noise.

The OCT parameter SRI showed a linear correlation with disease severity stage. Since there is a good structure–function correlation in RP [18], we can suggest that other SD-OCT parameters, in addition to EZ width, can be considered as criteria to determine disease stage and progression. However, Iftikhar et al. [14] found that in RP patients, there can be an apparent mismatch between structure and function in terms of the EZ and visual field. The visual field often extends significantly beyond the edges of their remaining EZ. They postulated that the EZ probably represents organized or densely packed photoreceptors and that there are fragmented photoreceptors beyond the discernible edges of the EZ that are alive and functioning. SD-OCT may not have a high enough resolution to detect such photoreceptors. Therefore, it is unreliable to depend on any single parameter and it is crucial to integrate multiple parameters for disease staging. According to Guymer et al. [13], retinal pigment epithelium (RPE) and outer retina (OR) atrophy (RORA) is defined on OCT as a region of signal hyper-transmission into the choroid, resulting from the interruption of the RPE and OR. Hyper-transmission could be considered as an indirect sign of atrophy [19]. Commercial OCT algorithms exist to quantify areas of increased choroidal signal due to RORA. In the past, RORA has been established as an anatomic biomarker to determine age-related macular degeneration stage and progression [14]. In this study, we evaluated the area of SRI. We decided to use SRI as a quantitative index of RPE and outer retina impairment. Our results indicated that the extent of RORA area, assessed by means of the parameter SRI, is correlated with RP severity score, and also with EZ score, suggesting that SRI can be considered as an additional predictive OCT biomarker.

Genotype might influence the progression of disease as a function of time. It may not be a coincidence that the lowest total score was achieved by patient 11 (F, 27 years old, CS: 1 in both eyes). Her below-average cumulative staging score could be due to the genetic variant found in one allele, an exon duplication (from exon 10 to 14). In genetics, deletions are considered prognostically more severe and are more often pathogenetic than duplications. Although a duplication constitutes a genomic rearrangement, it could represent a more favorable variant than a deletion in evaluating prognosis of RP. Indeed, it does not eliminate but rather adds information in the encoding protein, thus compensating for the total loss of function. Were such a hypothesis to prove true, it could also entail a “gain of function” effect that could be exploited for further therapeutic approaches. Clearly, in order to explore this hypothesis, further studies will be needed, extending the analysis to a cohort of patients with duplications on one allele, and comparing the progression of their disease with that of other patients without duplications.

## 5. Conclusions

The Iftikhar et al. RP severity score [14] was developed and evaluated on an analysis based on patients with typical RP of unknown genotype. Our findings add novel information, showing that in *USH2A*-related RP, the above severity score is significantly associated with age and with additional morpho-functional parameters not included in the staging scheme. This could prompt further validation studies to accurately determine the progression stage of this particular disease.

## Figures and Tables

**Figure 1 diagnostics-11-00213-f001:**
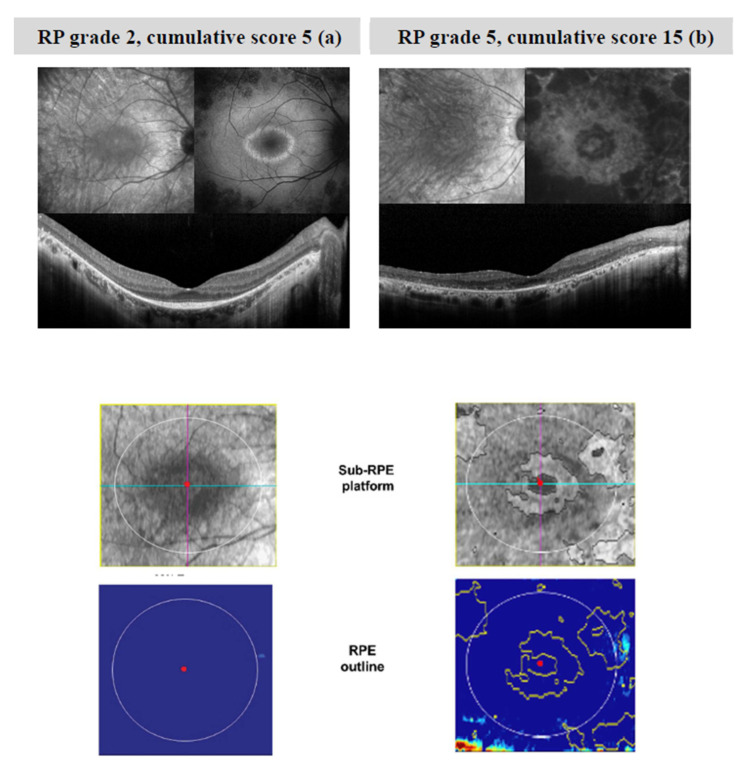
Images of infrared (IR), fundus autofluorescence (FAF), OCT B scan, sub-RPE platform and RPE outline of two patients with mild RP (**a**) and severe (**b**) RP. In mild RP (**a**), central preservation of the ellipsoid zone corresponds to the internal edges of the hyper-autofluorescent ring visible on FAF. In severe RP (**b**), the B scan image shows a profound loss of photoreceptor outer segments, with central loss of the RPE, corresponding to hypo-autofluorescent areas on FAF. This picture is associated with changes in the sub-RPE platform (areas of increased SRI) within the 5 mm circle outlined in white. The software provides an automated measure of the SRI area (the sum of areas outlined in yellow in the RPE outline). Patient (**a**) is 30 years old with a BCVA of 87 ETDRS letters (score 0 for BCVA), while patient (**b**) is 48 years old with a BCVA of 54 ETDRS letters (score 5 for BCVA). In patient (**a**), EZ extends for 9.21 degrees, whereas in patient (**b**), EZ is not recognizable on OCT scans. Thus, scores of 3 and 5 have been assigned to the EZ, respectively.

**Figure 2 diagnostics-11-00213-f002:**
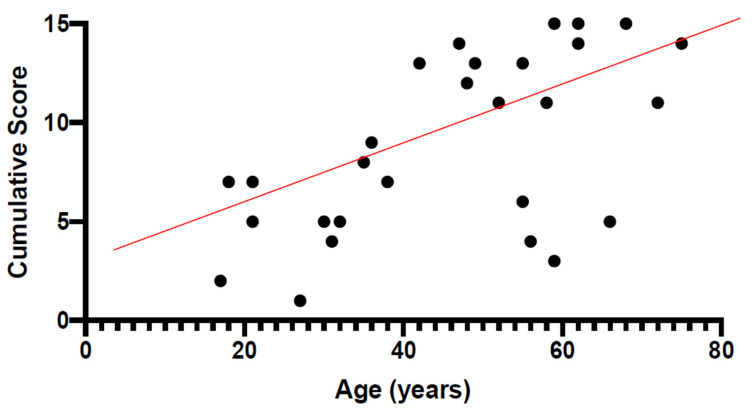
Cumulative score (CS) from the right eye of each *USH2A* patient plotted as a function of age. It can be noted that the score increases linearly with age, with a slope of 0.18/year. The r value is 0.65 (*p* < 0.01).

**Figure 3 diagnostics-11-00213-f003:**
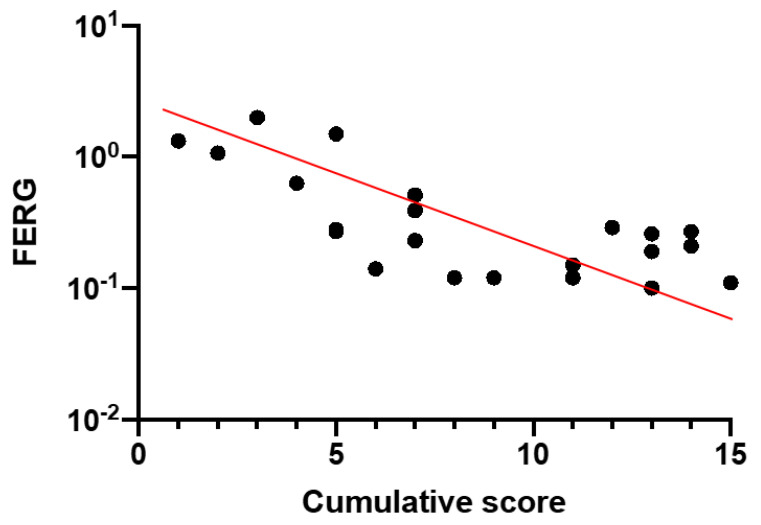
Focal Electroretinogram (FERG) Log 10 amplitudes plotted as a function of the CS (rho = −0.72, *p* < 0.0001).

**Figure 4 diagnostics-11-00213-f004:**
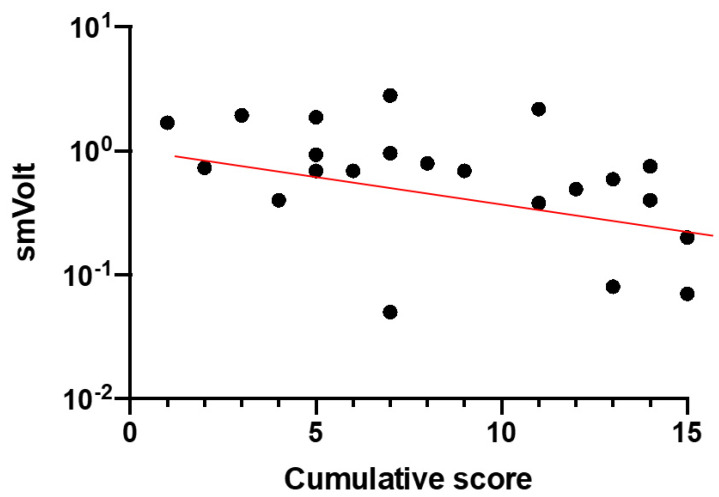
Log10 1F cone flicker microvolt ERG amplitude as a function of CS (rho = −0.58; *p* = 0.004).

**Figure 5 diagnostics-11-00213-f005:**
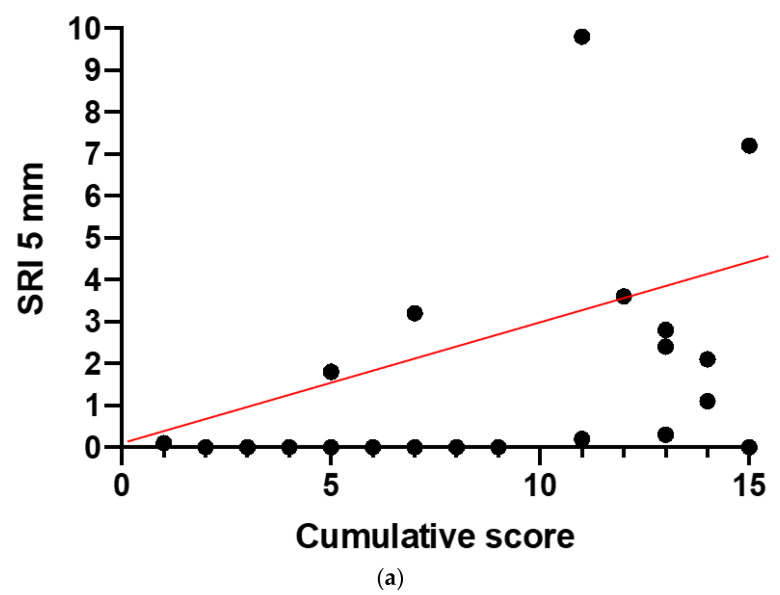
Results of correlation analysis between Sub-RPE Illumination (SRI) and CS (**a**) and Ellipsoid Zone (EZ) score (**b**). SRI showed a positive correlation with CS (*p* = 0.01) and the EZ score (*p* = 0.007).

**Table 1 diagnostics-11-00213-t001:** Genetic data.

	STATUS	Variant 1	ACMG	Variant 2	ACMG
**1**	HET	c.1031_1032del; p.(Phe344Cysfs*3)	P	c.6178dup; p.(Gln2060Profs*43)	P
**2**	HOM	c.9815C>T; p.(Pro3272Leu)	LP	c.9815C>T; p.(Pro3272Leu)	LP
**3**	HET	c.8584C>T; p.(Gln2862*)	P	c.14074G>A; p.(Gly4692Arg)	VUS
**4**	HET	c.15199del; p.(Ile5067Serfs*23)	P	c.10712C>T; p.(Thr3571Met)	P
**5**	HET	c.990_991del; p.(Asn330Lysfs*8)	P	c.10712C>T; p.(Thr3571Met)	P
**6**	HET	c.13130C>A; p.(Ser4377*)	P	c.653T>A; p.(Val218Glu)	LP
**7**	HOM	c.14248C>T; p.(Gln4750*)	P	c.14248C>T; p.(Gln4750*)	P
**8**	HET	c.10437G>T; p.(Trp3479Cys)	P	c.802G>A; p.(Gly268Arg)	P
**9**	HET	c.12067-2A>G; p.(?)	P	duplication in exons 10–14	VUS
**10**	HET	c.8584C>T; p.(Gln2862*)	P	c.13018G>C; p.(Gly4340Arg)	LP
**11**	HOM	c.10699del; p.(Leu3567*)	P	c.10699del; p.(Leu3567*)	P
**12**	HET	c.2276G>T; p.(Cys759Phe)	P	c.14286C>A; p.(Asn4762Lys)	VUS
**13**	HET	c.1055C>T; p.(Thr352Ile)	P	c.10712C>T; p.(Thr3571Met)	P
**14**	HET	c.13257_13263del; p.(Phe4419Leufs1*)	P	c.(2809+1_2810-1)_(2993+1_2994-1)del; p.(?)	P
**15**	HOM	c.5221T>C; p.(Ser1741Pro)	LP	c.5221T>C; p.(Ser1741Pro)	LP
**16**	HOM	c.2299del; p.(Glu767Serfs*21)	P	c.2299del; p.(Glu767Serfs*21)	P
**17**	HET	c.2299del; p.(Glu767Serfs*21)	P	c.4714C>T; p.(Leu1572Phe)	VUS
**18**	HET	c.1841-2A>G; p.(?)	P	c.10817T>C; p.(Leu3606Pro)	LP
**19**	HET	c.5386T>G; p.(Cys1796Gly)	LP	c.(14998_15131)_(15131_15393)del; p.(?)	P
**20**	HET	c.7501C>T; p.(Gln2501*)	P	c.908G>A; p.(Arg303His)	P
**21**	HOM	c.9815C>T; p.(Pro3272Leu)	LP	c.9815C>T; p.(Pro3272Leu)	LP
**22**	HET	c.13335_13347delinsCTTG; p.(Glu4445_Ser4449delinsAspLeu)	P	c.5153A>C; p.(Gln1718Pro)	VUS
**23**	HOM	c.9815C>T; p.(Pro3272Leu)	LP	c.9815C>T; p.(Pro3272Leu)	LP

ACMG: American College of Medical Genetics and Genomics classification. HET: Heterozygous, HOM: Homozygous, LP: likely pathogenetic, P: pathogenetic, VUS: variant of uncertain significance, * asterisk identifies the stop codon.

**Table 2 diagnostics-11-00213-t002:** Demographic and clinical data.

	NR	SEX	AGE	ONSET	RE					LE				
					BCVA	VF	EZ	CS	Grade	BCVA	VF	EZ	CS	Grade
*USH2A*	1	F	21	8	0	2	3	5	2	0	2	3	5	2
	2	F	59	16	5	5	5	15	5	4	5	5	14	5
	3	F	62	25	5	5	5	15	5	5	5	5	15	5
	4	M	18	13	1	1	5	7	3	1	1	5	7	3
	5	M	49	30	3	5	5	13	5	3	5	5	13	5
	6	F	21	19	3	0	4	7	3	1	0	4	5	2
	7	M	17	15	0	0	2	2	1	0	0	2	2	1
	8	M	48	18	3	5	4	12	4	3	5	5	13	5
	9	F	27	21	0	0	1	1	1	0	0	1	1	1
	10	M	32	22	0	0	5	5	2	0	1	5	6	2
	11	M	31	20	0	1	3	4	2	0	0	4	4	2
	12	M	55	54	0	1	5	6	2	0	1	5	6	2
	13	F	36	17	3	1	5	9	3	2	1	5	8	3
	14	M	62	20	4	5	5	14	5	4	5	5	14	5
	15	F	47	44	4	5	5	14	5	4	5	5	14	5
	16	F	30	23	0	3	2	5	2	0	2	2	4	2
	17	M	42	15	3	5	5	13	5	4	5	5	14	5
	18	M	55	25	5	3	5	13	5	4	4	5	13	5
	19	F	59	58	0	2	1	3	1	0	2	1	3	1
	20	M	72	43	4	2	5	11	4	4	1	5	10	4
NSRP	21	M	38	37	3	0	4	7	3	4	0	4	8	3
	22	F	52	20	3	3	5	11	4	3	3	5	11	4
	23	F	35	12	1	3	4	8	3	1	2	4	7	3

BCVA: best-corrected visual acuity; CS: cumulative score; EZ: ellipsoid zone; F: female; LE: left eye; M: male; N: number; NSRP: non-syndromic retinitis pigmentosa; RE: right eye; *USH2A*; Usher syndrome (USH) with variants in *USH2A* gene; VF: visual field.

**Table 3 diagnostics-11-00213-t003:** Demographic, clinical and morpho-functional data of the patients.

	N	Sex	Age	Onset	BCVA	VF	EZ	CS	Grade	SRI 5 mm Circle Area (mm^2^)	SRI Fovea Distance (mm)	Focal ERG (μV)	Submicrovolt 30 Hz (μV)
*USH2A*	1	F	21	8	0	2	3	5	2	0	0	0.28	0.93
	2	F	59	16	5	5	5	15	5	0	2.6	0.11	0.2
	3	F	62	25	5	5	5	15	5	7.2	0.3	0.11	0.07
	4	M	18	13	1	1	5	7	3	3.2	1.2	0.39	0.05
	5	M	49	30	3	5	5	13	5	0.3	1.6	0.26	0.08
	6	F	21	19	3	0	4	7	3	0	0	0.51	2.79
	7	M	17	15	0	0	2	2	1	0	0	1.07	0.73
	8	M	48	28	3	5	4	12	4	3.6	0.9	0.29	0.49
	9	F	27	21	0	0	1	1	1	0.1	2.3	1.32	1.68
	10	M	32	22	0	0	5	5	2	1.8	1.7	0.27	0.69
	11	M	31	20	0	1	3	4	2	0	2.6	0.63	0.4
	12	M	55	54	0	1	5	6	2	0	2.7	0.14	0.69
	13	F	36	17	3	1	5	9	3	0	3.6	0.12	0.69
	14	M	62	20	4	5	5	14	5	2.1	0.7	0.27	0.75
	15	F	47	44	4	5	5	14	5	1.1	1.1	0.21	0.4
	16	F	30	23	0	3	2	5	2	0	2.5	1.49	1.87
	17	M	42	15	3	5	5	13	5	2.4	0.4	0.19	0.59
	18	M	55	22	5	3	5	13	5	2.8	0.5	0.1	0.59
	19	F	59	58	0	2	1	3	1	0	0	1.99	1.93
	20	M	72	43	4	2	5	11	4	9.8	0.3	0.12	2.17
NSRP	21	M	38	37	3	0	4	7	3	0	0	0.23	0.96
	22	F	52	20	3	3	5	11	4	0.2	2	0.15	0.38
	23	F	35	12	1	3	4	8	3	0	3.5	0.12	0.79

BCVA: best-corrected visual acuity; CS: cumulative score; ERG: electroretinogram; EZ: ellipsoid zone; F: female; M: male; N: number; NSRP: non-syndromic retinitis pigmentosa; SRI: sub-RPE illumination; *USH2A*; Usher syndrome (USH) with variants in *USH2A* gene; VF: visual field.

**Table 4 diagnostics-11-00213-t004:** Results of the statistical analyses.

	Rho	*p*-Value
FERG vs. CS	−0.72	<0.0001
smVolt vs. CS	−0.58	0.004
SRI vs. CS	0.53	0.01
SRI vs. EZ	0.54	0.007

CS: Cumulative Score; EZ: ellipsoid zone; FERG: focal electroretinogram; smVolt: sub-microvolt 30 Hz; SRI: sub-RPE illumination.

## Data Availability

Data available from authors.

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
