# Peer review of "USH2A-Related Retinitis Pigmentosa: Staging of Disease Severity and Morpho-Functional Studies"

_diagnostics, 2021, doi:10.3390/diagnostics11020213_

Round 1

Reviewer 1 Report

Falsini et al reported that the correlations between the morphological and functional data of USH2A-related retinitis pigmentosa. The report would be of importance for preparing the future clinical study to treat the disease, however, there were several points to be clarified.

Although the definitions of cumulative score and grade may be reported previously, for the readers of this manuscript to easily understand, please briefly summarize the definitions.

The readers would like to know the age and visual function data of the patients in Figure 1.

The authors analyzed right eyes. Were there correlations between data of right and left eye data?

Approximate lines for Figures 3-5 would make the readers understand the results more easily.

Given that the disease grade is determined by cumulative score, the correlations between parameters, and cumulative score and grade would be parallel. Why did they think that it was meaningful to show the relationships both the parameters and the cumulative score, and the parameters and the grades?

They described that there were correlations between age and disease stage. Given that there were differences in onset, the progression speed would be different between the cases and the type of mutations. Why were there simple correlations between the age and disease stage? This would be of importance to understand the disease pathogenesis.

Line 256, the description, age-macular degeneration may be a typo.

Author Response

Reviewer 1

Falsini et al reported that the correlations between the morphological and functional data of USH2A-related retinitis pigmentosa. The report would be of importance for preparing the future clinical study to treat the disease, however, there were several points to be clarified.

Although the definitions of cumulative score and grade may be reported previously, for the readers of this manuscript to easily understand, please briefly summarize the definitions.

Thank you for your suggestion. We have added in the manuscript the definition of the cumulative score and grade.

“A scoring criterion was developed considering Early Treatment Diabetic Retinopathy Study (ETDRS) visual acuity, Goldmann visual field area and EZ width and by assigning a score from 0 to 5 to each variable. Normal values correspond to a score of 0 (85 ETDRS or better for visual acuity, 120° or better for visual field diameter, and 30° or better for ellipsoid zone width), whereas a score of 1–5 is assigned based on the population quintiles [14]. The cumulative score (CS, from 0 to 15) is represented by the sum of all scores and it is used to determine the severity grade from 0 to 5 ” (from line 109 to line 114).

The readers would like to know the age and visual function data of the patients in Figure 1.

Thank you for your suggestion. The age and visual function of the patients in Figure 1 have been added in Figure caption. Patient (A) is 30 years old with a BCVA of 87 ETDRS letters (score 0 for BCVA), while patient (B) is 48 years old with a BCVA of 56 ETDRS letters (score 4 for BCVA). 

The authors analyzed right eyes. Were there correlations between data of right and left eye data?

We analyzed both right eyes and left eyes and the results from the statistical analyses were substantially similar. In addition, data from right and left eyes were highly correlated (cumulative score inter-eye correlation: r = 0.98).  In the study, we considered only the results from the right eyes in order not to overestimate the p values.

(we have added these comments to the “statistical analyses” section - line 137-138 - and to the “results” section - line 164 to 167- ).

Approximate lines for Figures 3-5 would make the readers understand the results more easily.

Thank you. Approximate lines have been added to the figures 3-5.

Given that the disease grade is determined by cumulative score, the correlations between parameters, and cumulative score and grade would be parallel. Why did they think that it was meaningful to show the relationships both the parameters and the cumulative score, and the parameters and the grades?

Thank you. Although we can assume that cumulative score and grade are parallel, the correlation has to be verified. According to the Reviewer suggestion, we removed the correlations between parameters and grades.

They described that there were correlations between age and disease stage. Given that there were differences in onset, the progression speed would be different between the cases and the type of mutations. Why were there simple correlations between the age and disease stage? This would be of importance to understand the disease pathogenesis.

Thank you for your comment. This point is very interesting. We suggest that for USH2A patients the age of onset tends to be quite homogeneous. Therefore, the cumulative severity score results to be positively correlated with patient age (this consideration has been added in the “discussion” section, line 371 to 373).

Line 256, the description, age-macular degeneration may be a typo.

Thank you. We modified “age-related macular degeneration”.

Reviewer 2 Report

This manuscript describes the authors’ patient cohort with RP associated with mutations in the USH2A gene. Although it is indeed important to study RP patients caused by mutations in the single gene, this reviewer thinks that there are scientifically serious problems in the analytical methods in this study design. The major points are described below.

  1. Explanations are needed on the method how the authors defined the “ellipsoid zone” on SD-OCT pictures. This reviewer points out examples using the patients illustrated in Fig. 2 a and b. In the case of Fig. 2a, there is a single diffuse hyperreflective zone instead of a discrete EZ in the photoreceptor inner and outer segment area. In addition, no EZ can be identified in the patient of Fig. 2b. This reviewer suspects that it is sometimes difficult to identify the EZ in some patients with RP whose photoreceptors are more severely damaged than a certain level.

  1. Statistical analyses

It is not suitable to apply either Pearson’s or Spearman’s correlation analysis using non-continuous variables such as cumulative severity score or severity grade. Furthermore, it is not suitable to use Pearson’s correlation analysis using FERG and smVolt values, because these values are not shown to be distributed according to a normal distribution (the reviewer’s analysis using Shapiro-Wilk test, P < 0.001, P = 0.003, respectively). The authors need to reconsider the rationale that allows them to analyze the relationship between the severity score or grade and other parameters using statistical methods to characterize the severity of their patients with RP associated with mutations in the USH2A. However, this reviewer still thinks that under a special limitation that cumulative severity score is considered as nearly equal to a kind of continuous variables, the correlation between the cumulative severity score and age is meaningful for this particular patient-cohort. Because both variables show a normal distribution (the reviewer’s analysis using Shapiro-Wilk test, P = 0.141 and P = 0.434, respectively), they can be statistically authorized to be analyzed by Pearson’s correlation analysis.

Author Response

Reviewer 2

This manuscript describes the authors’ patient cohort with RP associated with mutations in the USH2A gene. Although it is indeed important to study RP patients caused by mutations in the single gene, this reviewer thinks that there are scientifically serious problems in the analytical methods in this study design. The major points are described below.

  1. Explanations are needed on the method how the authors defined the “ellipsoid zone” on SD-OCT pictures. This reviewer points out examples using the patients illustrated in Fig. 2 a and b. In the case of Fig. 2a, there is a single diffuse hyperreflective zone instead of a discrete EZ in the photoreceptor inner and outer segment area. In addition, no EZ can be identified in the patient of Fig. 2b. This reviewer suspects that it is sometimes difficult to identify the EZ in some patients with RP whose photoreceptors are more severely damaged than a certain level.

Thank you for your comment. On OCT line scans, two boundaries were identified: the proximal edge of the retinal pigment epithelium (pRPE) located adjacent to the photoreceptor outer segments and the EZ band. For all scans, the nasal and temporal edges of the EZ band were defined as the locations where the EZ band met the pRPE. The width of the EZ band was defined as the distance between these two locations (the explanation has been included in the “methods” section, from line 127 to line 131).

EZ is well recognizable in Patient (a), Figure 1. However, it can often be difficult to identify EZ in RP patient, as shown in Patient (b), Figure 1. As suggested by Iftikhar et al. [14], there is an apparent mismatch between structure and function in terms of the EZ and visual field in RP patients.  The visual field often extends significantly beyond the edges of their remaining EZ. They postulated that the EZ probably represents organized or densely packed photoreceptors and that there are fragmented photoreceptors beyond the discernible edges of the EZ that are alive and functioning. SD-OCT may not have a high enough resolution to detect such photoreceptors. Therefore, it is unreliable to depend on any single parameter and it is crucial to integrate multiple parameters for disease staging (added to the manuscript in “discussion section”, from line 399 to line 405).  

  1. Statistical analyses

It is not suitable to apply either Pearson’s or Spearman’s correlation analysis using non-continuous variables such as cumulative severity score or severity grade. Furthermore, it is not suitable to use Pearson’s correlation analysis using FERG and smVolt values, because these values are not shown to be distributed according to a normal distribution (the reviewer’s analysis using Shapiro-Wilk test, P < 0.001, P = 0.003, respectively). The authors need to reconsider the rationale that allows them to analyze the relationship between the severity score or grade and other parameters using statistical methods to characterize the severity of their patients with RP associated with mutations in the USH2A. However, this reviewer still thinks that under a special limitation that cumulative severity score is considered as nearly equal to a kind of continuous variables, the correlation between the cumulative severity score and age is meaningful for this particular patient-cohort. Because both variables show a normal distribution (the reviewer’s analysis using Shapiro-Wilk test, P = 0.141 and P = 0.434, respectively), they can be statistically authorized to be analyzed by Pearson’s correlation analysis.

According to the Reviewer’s remark we re-evaluated the correlations between FERG and SmVolt values and cumulative score, by using Spearman’s correlation analysis. The results were highly significant and are now reported in the Results section (line 205-206, line 210-211, 256-257).

Reviewer 3 Report

I consider this manuscript to have valable data that would be of readers' interest if published. As mentioned in the paper USH2A-related Retinsitis Pigmentosa is  a rare disease which result in uncorectable blindness but the new thearpies have very promising results. So this is very important to clasify propeprly the eye changes. I have only minor remark to the present study: please show the Figure 5 statistical analyses in a Table. 

Author Response

Reviewer 3

I consider this manuscript to have valable data that would be of readers' interest if published. As mentioned in the paper USH2A-related Retinsitis Pigmentosa is  a rare disease which result in uncorectable blindness but the new thearpies have very promising results. So this is very important to clasify propeprly the eye changes. I have only minor remark to the present study: please show the Figure 5 statistical analyses in a Table. 

Thank you for your comment. As you’ve suggested, we’ve summarized the results of statistical analyses in Table n. 4. The Table has been added to the manuscript (line 325-340).  

rho

P value

FERG vs Cumulative Score

-0.72

<0.0001

smVolt vs  Cumulative Score

-0.58

0.004

SRI vs

Cumulative Score

0.53

0.01

SRI vs EZ

0.54

0.007

Round 2

Reviewer 2 Report

The authors have fairly well revised the manuscript. However, the description seems to still insufficiently respond to the previous comments that this reviewer had raised.

  1. The authors need to clarify how they defined EZ band in patients who had not shown any EZ band on SD-OCT montage such as Fig. 1b.
  2. The authors need to elaborate on the rationale why they performed Spearman’s correlation analysis using non-continuous variables such as cumulative severity score or severity grade. This is a simple issue of statistics.

Author Response

Manuscript ID: diagnostics-1059914

USH2A-related Retinitis Pigmentosa: staging of disease severity and morpho-functional studies.

Response to the reviewers, 2nd round

Dear Editor

We would like to thank again the Reviewers for their constructive comments. The comments are addressed in a point-by-point fashion below (in red).

Reviewer 2

The authors have fairly well revised the manuscript. However, the description seems to still insufficiently respond to the previous comments that this reviewer had raised.

  1. The authors need to clarify how they defined EZ band in patients who had not shown any EZ band on SD-OCT montage such as Fig. 1b.

In patient shown in Figure 1 b, EZ is not identifiable on OCT scans. Therefore, the patient has been assigned a value of zero degrees, which corresponds to a score of 5 for the EZ.

In patient shown in Figure 1 a, EZ extends for 9.21 degrees. Thus, a score of 3 has been assigned to the EZ. (EZ scores have been added to the Figure 1 caption).

  1. The authors need to elaborate on the rationale why they performed Spearman’s correlation analysis using non-continuous variables such as cumulative severity score or severity grade. This is a simple issue of statistics.

We modified the test from line 188 to line 190. “For non-continuous ordinal variables such as cumulative severity score, and given the non-gaussian data distribution, the non-parametric Spearman rank order correlation analysis was considered more appropriate than Pearson’s correlation”.